# Susceptibility of the Formate Hydrogenlyase Reaction to the Protonophore CCCP Depends on the Total Hydrogenase Composition

**Janik Telleria Marloth and Constanze Pinske *** 

Institute for Biology/Microbiology, Martin-Luther-University Halle-Wittenberg, 06108 Halle, Germany; janik.telleria-marloth@student.uni-halle.de

**\*** Correspondence: constanze.pinske@mikrobiologie.uni-halle.de; Tel.: +49-(0)-345-5526-353

**Abstract:** Fermentative hydrogen production by enterobacteria derives from the activity of the formate hydrogenlyase (FHL) complex, which couples formate oxidation to $H_2$ production. The molybdenum-containing formate dehydrogenase and type-4 [NiFe]-hydrogenase together with three iron-sulfur proteins form the soluble domain, which is attached to the membrane by two integral membrane subunits. The FHL complex is phylogenetically related to respiratory complex I, and it is suspected that it has a role in energy conservation similar to the proton-pumping activity of complex I. We monitored the $H_2$-producing activity of FHL in the presence of different concentrations of the protonophore CCCP. We found an inhibition with an apparent $EC_{50}$ of 31 μM CCCP in the presence of glucose, a higher tolerance towards CCCP when only the oxidizing hydrogenase Hyd-1 was present, but a higher sensitivity when only Hyd-2 was present. The presence of 200 mM monovalent cations reduced the FHL activity by more than 20%. The $Na^+/H^+$ antiporter inhibitor 5-(*N*-ethyl-*N*-isopropyl)-amiloride (EIPA) combined with CCCP completely inhibited $H_2$ production. These results indicate a coupling not only between $Na^+$ transport activity and $H_2$ production activity, but also between the FHL reaction, proton import and cation export.

**Keywords:** formate hydrogenlyase; hydrogen metabolism; energy conservation; MRP (multiple resistance and pH)-type $Na^+/H^+$ antiporter; CCCP—carbonyl cyanide *m*-chlorophenyl-hydrazone; EIPA—5-(*N*-ethyl-*N*-isopropyl)-amiloride

## 1. Introduction

Anaerobic or fermentative growth in the absence of oxygen requires that conservation of energy is at its utmost efficiency. The reaction of the formate hydrogenlyase (FHL) complex contributes indirectly to the generation of a proton gradient at the cytoplasmic membrane by consumption of protons ($H^+$) from the cytoplasm, diffusion of $H_2$-gas across the membrane and subsequent oxidation by the two periplasmic $H_2$-oxidizing hydrogenases (Hyd-1 and Hyd-2) [1] (Figure 1). Hyd-1 forms a redox loop, and Hyd-2 contributes directly to proton transfer from the cytoplasm to the quinone during oxidation of $H_2$ [1,2]. The *hyaB* and *hybC* genes encode the catalytic subunits of Hyd-1 and Hyd-2, respectively [3]. This kind of intracellular syntrophy was initially described for *Desulfovibrio* species [4] and more recently for *Acetobacterium woodii* [5]. The FHL complex of *Escherichia coli* is active during mixed acid fermentation to disproportionate formate, which, when accumulated intracellularly, has cytotoxic effects [6]. The two active sites of the FHL complex comprise the [NiFe]-hydrogenase 3 (HycE protein) and the molybdenum- and selenium-containing formate dehydrogenase H (FdhH protein), which function together with three electron-transferring iron-sulfur carrying subunits located in the cytoplasm. Membrane attachment to the membrane-integral HycC and HycD subunits is required for

coupling of these half-reactions [7,8]. Our previous studies have shown that the FHL complex can work in reverse in resting cells, a reaction of great biotechnological interest for $H_2$ and $CO_2$ storage as formate [8,9]. A second FHL complex with a predicted five instead of two membrane subunits is encoded in the *hyf*-operon on the *E. coli* chromosome, but, presumably due to its low level of gene expression, the activity of this enzyme complex is not detectable under our growth conditions [10,11].

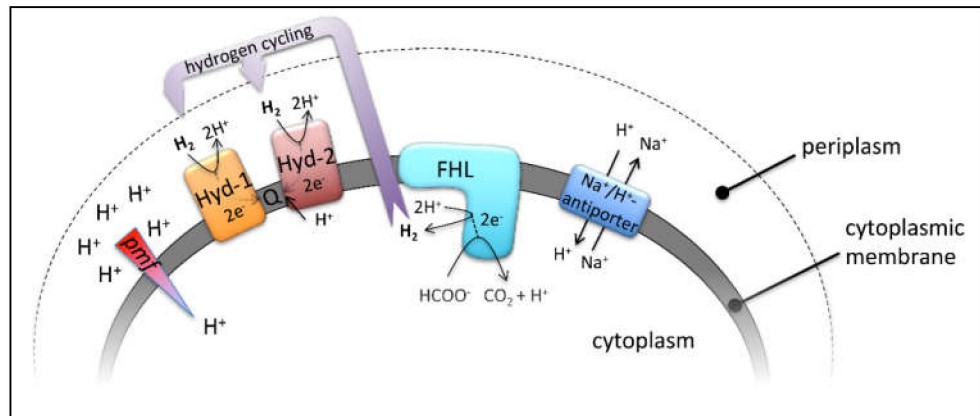

**Figure 1.** Relevant enzyme complexes of this study, their reactions and their membrane localization. Hyd-1 and Hyd-2 indicate the periplasmic $H_2$-oxidizing hydrogenases, which channel electrons into the quinone pool (Q), FHL—the cytoplasmic formate hydrogenlyase—and the membrane bound $Na^+/H^+$ antiporters. Further details are highlighted in the main text.

When the phylogenetic relationship between the FHL complex and complex I was discovered, energy conservation by proton-pumping of the complex was also assumed [12,13]. The hydrogenase 3 (HycE protein) belongs to the class 4 hydrogenases and is similar to the Ech (Energy-converting hydrogenase) hydrogenases, which were shown to generate proton gradients during catalysis [14]. However, while studying FHL bidirectionality as the basis for analyzing the influence of ionophores, we unexpectedly found that the protonophore carbonyl cyanide *m*-chlorophenyl-hydrazone (CCCP) showed only a 40% inhibition of the $H_2$ production activity and a 30% increase of the reverse formate production reaction from $H_2$ and $CO_2$ [8].

It was speculated that due to the small difference in standard redox potentials of formate ($E_0' = -432$ mV, pH 7.0) and $H_2$ ($E_0' = -414$ mV, pH 7.0), redox energy for proton translocation becomes available only when the physiological conditions shift to a more acidic pH [15]. However, the reversibility of an artificially coupled hydrogenase and formate dehydrogenase reaction from *Desulfovibrio* via an electrode was able to mimic the FHL reaction in both directions in vitro without dependence of the reaction on energy provision [16]. Instead, a controversially discussed model of a multienzyme transport supercomplex comprising $F_1F_0$-ATPase, the $K^+$-transporting TrkA symporter and the FHL(-2) complex was suggested for energetic coupling [17,18]; however, its existence still requires verification (reviewed in [19]). Therefore, both the necessity for and the role of the FHL complex's membrane attachment in energy conservation remains to be resolved.

In the present work we have systematically re-examined the CCCP effect on FHL activity, whereby we combined different CCCP concentrations with use of different carbon sources to initiate FHL activity in strains either bearing or lacking the $H_2$-oxidising Hyds. We found that ultimately all strains could be inhibited by CCCP, but strains lacking Hyd-1 required lower, and those lacking Hyd-2 required higher, concentrations of CCCP for inhibition. Based on the relationship of the membrane subunit HycC to $Na^+/H^+$ (MRP)-type antiporters, we further addressed the question of whether the FHL reaction is driven by or is contributing to the *pmf* and if it might be linked to the sodium motive force. We present strong evidence for a dependence of the FHL reaction on proton influx and sodium export.

## 2. Results and Discussion

*2.1. CCCP Inhibits H$_2$ Production to Different Degrees Depending on the Hydrogenase Composition*

Cell suspensions initiate H$_2$ production after the anaerobic addition of either formate or glucose, the latter requires metabolizing by the cells to formate prior to H$_2$ production. Consequently, the mixed-acid fermentation yields 2 mol formate per mol glucose (reviewed in [3]). Formate is the direct substrate for the FHL complex, but its import via the formate channel FocA could mask the effects seen. FocA behaves as a pH-dependent channel in vitro, while its activity is mostly determined by its interaction with the pyruvate formate lyase protein PflB in vivo [20]. It is feasible that formate export across the membrane contributes to *pmf* generation by proton symport, but the import mechanism is unresolved, except that it only occurs when formate-consuming reactions in the cytoplasm can take place [21]. Therefore, the apparent 1.3 mol H$^+$ that are translocated across the membrane per mole of formate oxidized during the FHL reaction could be indirect [22]. On the other hand, the transport of glucose across the membrane is PTS (phosphotransferase system)-dependent, where phosphoenolpyruvate provides the phosphoryl group for the incoming glucose in a *pmf*-independent manner [23]. Our parallel experimental design circumvents the fact that indirect effects through the transport of the carbon source into the cells are observed.

The parental strain MC4100, the isogenic *hyaB* deletion strain CP630, which lacks Hyd-1 activity, the isogenic *hybC* deletion strain CP631, which lacks Hyd-2 activity, or the *hyaB hybC* double deletion strain CP734 were grown under the optimal conditions for FHL activity. Subsequently, cells were harvested and suspensions used on a modified Clark-type electrode to monitor H$_2$ production. The strains showed initial activities very similar to each other in the presence of the same carbon source, but generally higher activity was observed in the presence of formate than glucose (Table 1). A strain lacking the genes for the FHL complex (HDK103) did not produce H$_2$ under these conditions, supporting the observation that the second FHL is inactive under the conditions tested here [10].

**Table 1.** H$_2$ production rates in nmol H$_2$·min$^{-1}$·mg$^{-1}$ ± standard deviation from at least three independent biological samples.

| Strain [1] | Glucose | Formate |
|---|---|---|
| MC4100 | 69 ± 12 | 160 ± 47 |
| CP630 (Δ*hyaB*) | 74 ± 23 | 173 ± 13 |
| CP631 (Δ*hybC*) | 101 ± 15 | 145 ± 20 |
| CP734 (Δ*hyaB* Δ*hybC*) | 114 ± 24 | 121 ± 13 |

[1] Strains were grown in TGYEP, pH 6.5 medium for 16 h at 30 °C, cells harvested and suspended in 50 mM Tris/HCl, pH 7.0, prepared and assayed on the Clark-type electrode.

When CCCP concentrations of 100 µM were employed, the effect this had on the H$_2$ production was strain dependent. A previous study from our lab had shown that 100 µM CCCP caused a slight inhibition of the H$_2$ production rate in a strain with a similar Hyd composition to CP734 [8]. If indeed proton translocation is directly performed by the FHL complex, then CCCP would be expected to cause an increase of the activity, not a decrease. Nevertheless, 100 µM CCCP caused complete inhibition of H$_2$ production by the parental strain MC4100 independent of the employed carbon source. The slope of the H$_2$ production even inverted (i.e., H$_2$ oxidation) after CCCP addition (Figure 2), which indicates consumption of the produced H$_2$ within the electrode system and indicates the activity of H$_2$-oxidation reactions, which are usually not performed by the FHL complex, but by the H$_2$-oxidizing Hyd-1 and Hyd-2 enzymes, indicating they are responsible for the difference.

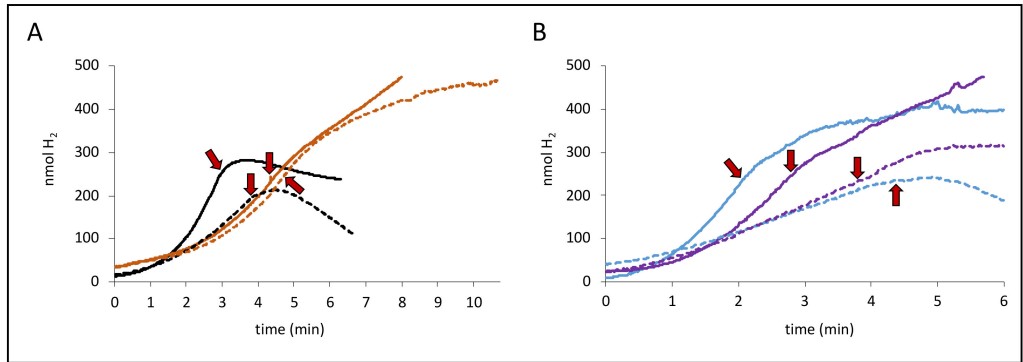

**Figure 2.** Different degrees of inhibition by protonophore carbonyl cyanide *m*-chlorophenyl-hydrazone (CCCP). Panel A shows $H_2$ production from the strains MC4100 (parental, black curves) and CP734 (*ΔhyaB ΔhybC*, orange curves) while panel B shows the strains CP630 (*ΔhyaB*, blue curves) and CP631 (*ΔhybC*, purple curves) after addition of either 14 mM glucose (dashed lines) or 50 mM formate (solid lines). When $H_2$ production was linear, a final concentration of 100 μM CCCP was added to the reaction vessel, as indicated by the red arrows.

In the next step, strains lacking the genes encoding the large subunit of either Hyd-1 (*ΔhyaB*, strain CP630) or Hyd-2 (*ΔhybC*, strain CP631) were used in the same experimental setup. While the Hyd-2-deficient strain CP631 (purple curves) mimics the $H_2$ production curves of the double deletion strain CP734 (*ΔhyaB ΔhybC*) after CCCP addition, the Hyd-1 deficient strain CP630 (blue curves) looks very similar to the parental curves and negative slopes after CCCP addition is observed. It can be further observed that the effect of CCCP addition is not immediate, but $H_2$ production rates shift rather gradually. For the direct dependence of an activity on *pmf*, we have previously observed immediate and complete effects of CCCP addition on the activity [2], suggesting an indirect effect of CCCP occurs here. Notably, the addition of other protonophores like 2,4-dinitrophenol (DNP) up to 1 mM had no effect on the $H_2$ production activity of the cells, similar to what was observed before at lower concentrations [8]. Taken together, this indicates that the effect of CCCP on FHL activity is indirect.

*2.2. Hyd-1 Confers Resistance and Hyd-2 Sensitivity of $H_2$ Production to CCCP*

In order to quantify the effect of CCCP on the FHL activity, we calculated the apparent $EC_{50}$ (half maximal effective concentration) values (Table 2). For that purpose, we analyzed the $H_2$ production on the electrode, and at about 100 nmol $H_2$, different concentrations of CCCP were added to the reaction. Figure 3A shows an example of the effect of the CCCP addition on strain CP630 (*ΔhyaB*). A concentration of 1 and 10 μM did not significantly alter the slope of $H_2$ production, while 50 and 100 μM CCCP clearly slowed down the reaction, and 200 μM led to complete stagnation of $H_2$ production. A further increase to 400 μM CCCP revealed negative slopes, indicating re-oxidation of $H_2$ by Hyd-2. Due to slight variations in the initial FHL activities, the reduction of activity after CCCP addition was calculated as the ratio of the activity after CCCP addition to the activity before its addition (Figure 3B). In the case of a negative slope, the ratio after CCCP addition was calculated as zero.

**Table 2.** Apparent $EC_{50}$ values of CCCP inhibition.

| Strain [1] | Glucose | Formate |
|---|---|---|
| MC4100 | 31 μM | 81 μM |
| CP630 (*ΔhyaB*) | 0.13 μM | 42 μM |
| CP631 (*ΔhybC*) | 125 μM | 223 μM |
| CP734 (*ΔhyaB ΔhybC*) | 98 μM | 348 μM |

[1] The same cell suspensions were used for the glucose and formate experiments. The same amount of total cell protein (1 mg) was used for the different strains. All values were calculated from at least five different CCCP concentrations.

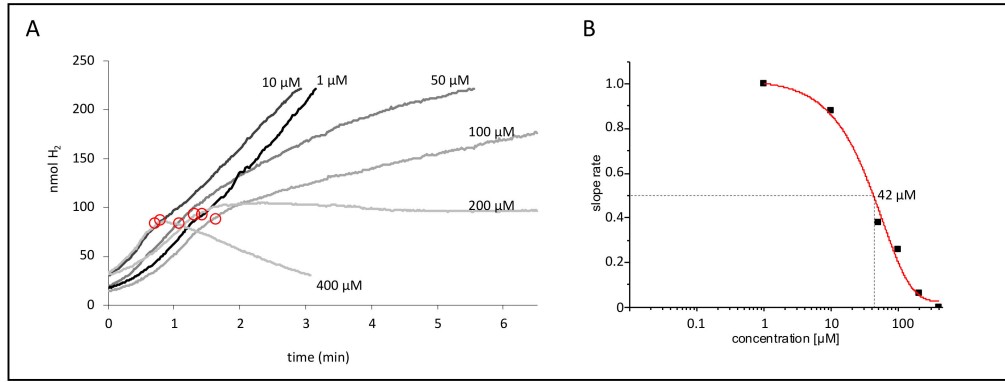

**Figure 3.** Dose-dependent effect of CCCP. (**A**) Strain CP630 (Δ*hyaB*) was used as a representative strain for CCCP inhibition after initiation of the reaction with 50 mM formate and linear slope; the designated concentrations of CCCP were added at the time points encircled in red. (**B**) The apparent $EC_{50}$ values of this set of activities were calculated from the ratio of slopes after and before CCCP addition and subsequent fitting with the $EC_{50}$ function in SigmaPlot.

An apparent half-maximal inhibition ($EC_{50}$) of $H_2$ production can be observed at 31 μM CCCP with glucose or 81 μM CCCP with formate as substrate in strain MC4100 (Table 2). These higher concentrations of CCCP, which are required in the presence of formate compared to glucose, reflect the higher initial activities in the presence of formate (Table 1). The double deletion strain CP734 (Δ*hyaB* Δ*hybC*) only showed reduction of $H_2$ production at elevated CCCP concentrations, which is reflected in the increased apparent $EC_{50}$ values. For example, we observed complete inhibition of MC4100 (glucose initiated) at 50 μM CCCP but in strain CP734 only at 500 μM CCCP. Looking at the effect of single deletions of the two $H_2$-oxidizing Hyd in strains CP630 and CP631, respectively, then the Hyd-1 deletion (Δ*hyaB*, strain CP630) caused increased sensitivity to CCCP compared to the parental MC4100, while the Hyd-2 deletion (Δ*hybC*, strain CP631) increased resistance to CCCP similar to the double deletion strain CP734 (Table 2).

Hyd-1 has a lower catalytic activity than Hyd-2 [24,25], while at the same time it has a similar abundance in the membrane under the conditions used to grow the cells here [26]. It is possible that the inactivation of Hyd-1 in strain CP630 results in elevated amounts of Hyd-2 because more maturation enzymes for cofactor construction and more membrane space are available. Hyd-2 is the enzyme that contributes to *pmf* generation [27], and the data show that Hyd-2 but not Hyd-1 catalyzes the oxidation of $H_2$ at high CCCP concentrations. CCCP enhances Hyd-2 activity by removing the back-pressure of the *pmf* on the enzyme. When assayed directly, the effect is not very prominent because the enzyme works at its catalytic optimum, but in variants that exhibit difficulties in transferring electrons across the membrane, the effect is more pronounced [2]. Electrons from Hyd-2 are concomitantly channeled into the quinone pool for reduction of electron acceptors like fumarate, which can be internally produced during mixed-acid fermentation when glucose is provided to the cells [28]. Therefore, negative slopes occur mostly when Hyd-2 is present and glucose is provided. This does not account for $H_2$ oxidation in the presence of formate; however, we cannot currently explain this effect (Figure 3A). The higher amounts of Hyd-2 enzyme in combination with its CCCP-elevated activity account for the apparent higher CCCP sensitivity in the absence of Hyd-1. Therefore, the absence of Hyd-2 confers resistance, and the absence of Hyd-1, resulting in a concomitant increase in Hyd-2 levels, causes sensitivity to CCCP.

CCCP is, however, also effective at stalling the $H_2$ production when Hyd-1 and Hyd-2 are both absent, albeit only at higher concentrations, but independent of the given carbon source. Hence, a secondary effect, exclusively based on FHL activity, must occur. When, in an unrelated study, a proteorhodopsin was expressed in *E. coli* cells, the $H_2$ production from FHL was increased upon light exposure, which initiates proton-pumping activity of the proteorhodopsin [29] and supports the

observed *pmf*-consumption during H$_2$-evolution observed here. We can exclude a direct inhibition of either the hydrogenase or formate dehydrogenase half-reaction of the FHL complex. When assayed independently of one another with benzyl viologen redox dye, crude extracts from strain CP734 showed a hydrogenase activity of 2.43 ± 0.48 U mg$^{-1}$ in the absence and 2.36 ± 0.42 U mg$^{-1}$ in the presence of CCCP. The formate dehydrogenase activity was similarly unaltered with an activity of 0.26 ± 0.09 U mg$^{-1}$ without and 0.27 ± 0.07 U mg$^{-1}$ with CCCP present.

## 2.3. CCCP Promotes the Reverse FHL Reaction

Due to the inhibitory effect of CCCP addition on the H$_2$ production of FHL, we investigated its effect on the reverse FHL reaction, which can be initiated if resting cells are incubated under elevated pressure of H$_2$ and CO$_2$ gas mix, as exploited before [8,9]. The reverse reaction was tested in the presence of different concentrations of CCCP, and the resulting formate concentration secreted by the cell suspensions at the equilibrium state was subsequently analyzed using HPLC (Figure 4). While cells produced 23 mM formate in the absence of CCCP in a strain that lacked Hyd-1 and Hyd-2, no formate formation was observed in the parental strain MC4100 under these conditions. The amount of formate initially decreased in the presence of 1, 10 and 50 µM CCCP, was then similar to no addition in the presence of 75 µM CCCP and finally increased to 180 mM formate when 500 µM CCCP was added. The parental strain MC4100 only showed formate production when 200 µM CCCP or more was present but eventually produced 157 mM formate in the presence of 500 µM CCCP (Figure 4). This difference between the two strains clearly showed that when Hyd-1 and Hyd-2 are present (strain MC4100), H$_2$ oxidation by these enzymes is the preferred route for H$_2$ oxidation to the FHL reverse reaction. Generally, an increase of the CCCP concentration resulted in an increase of formate production. If the CCCP effect was *pmf*-mediated, then this could be interpreted as CCCP-mediated relief of the *pmf* counter-pressure during the reverse FHL reaction. An alternative explanation is provided by the proton symport of the formate channel FocA during export, which would be enhanced in the presence of CCCP, thus removing cytoplasmic formate from the reaction and, therefore, shifting the equilibrium of the FHL reaction towards formate production.

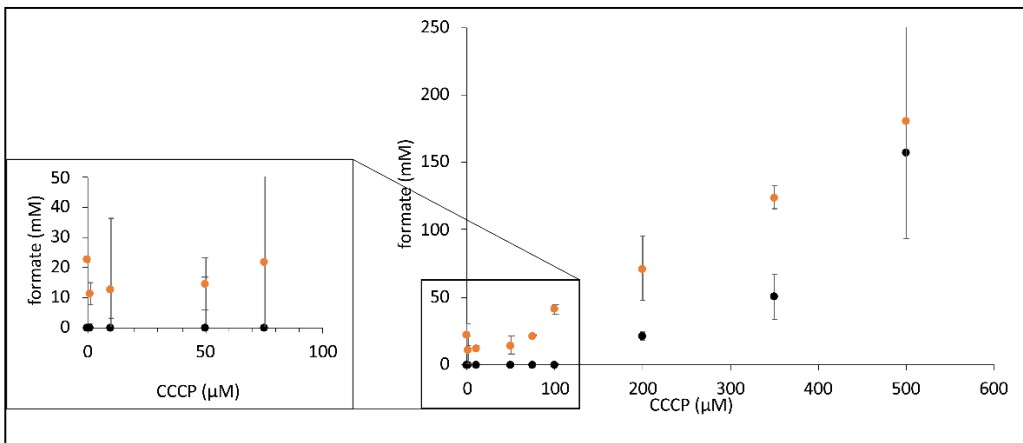

**Figure 4.** CCCP-dependent increase in formate synthesis by FHL. Strains MC4100 (parental, black dots) and CP734 (Δ*hyaB* Δ*hybC*, orange dots) were triggered for formate production in the presence of different concentration of CCCP as indicated (described in the Materials and Methods section). Each dot represents the average of three biological replicates with standard deviation.

## 2.4. The DTT Reversal of the CCCP Effect Is Not Mediated by Hyd-2 Activity

It has been observed that the effects of CCCP can be generally relieved by addition of dithiothreitol (DTT) [30] and also specifically restore the activity of FHL with DTT after CCCP inhibition [17]. Initially, it was verified that the addition of DTT on its own had no effect on the H$_2$ production rate of MC4100 (Figure 5A). Once the H$_2$ production was CCCP-inhibited, the addition of DTT was able to gradually

restore $H_2$ production up to the rate it had been before CCCP addition (Figure 5A). We also observed that the DTT relief of CCCP inhibition is independent of the strain used, indicating that the presence of Hyd-2 or Hyd-1 is not essential for the reversibility. Furthermore, CCCP was unable to inhibit $H_2$ production when it was already pre-incubated with DTT (red curve in Figure 5A).

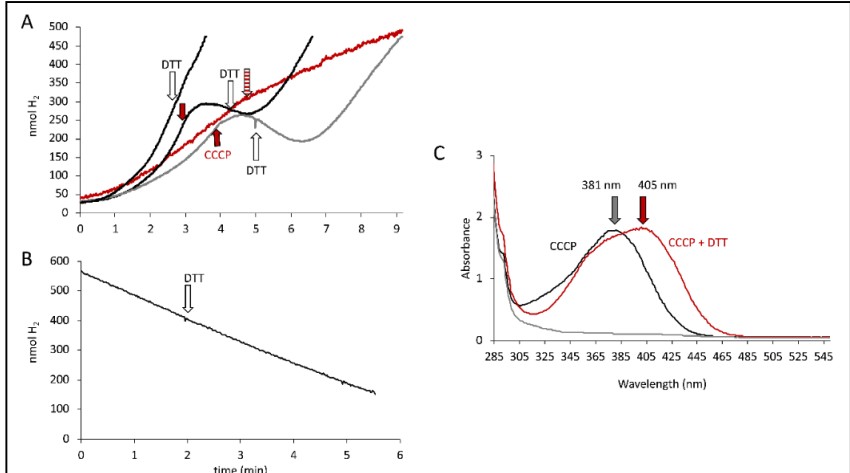

**Figure 5.** DTT effect. (**A**) Electrode traces of MC4100 cells producing $H_2$ from either glucose (grey curves) or formate (black curves). CCCP addition (red arrow, 100 μM) and DTT addition (white arrow, 5 mM) are indicated. The red trace shows the combined addition of DTT and CCCP (striped red arrow). (**B**) Electrode trace of strain HDK103 (Δ*hyaB* Δ*hycA-I*) showing the activity of Hyd-2 as $H_2$ oxidation of $H_2$-saturated buffer after the addition of 15 mM fumarate. The white arrow indicates the addition of 5 mM DTT. (**C**) Absorption spectra of CCCP solution (black), CCCP solution after DTT addition (red) and DTT alone (grey). The arrows indicate the respective maxima.

We were recently able to show that Hyd-2 contributes to *pmf* generation in an electrogenic fashion that is susceptible to CCCP addition [2,27]. If CCCP addition substantially increased Hyd-2 activity to the amount where it outcompeted the $H_2$ production of the FHL complex and thus reduced the overall $H_2$ production, then the DTT addition would inhibit Hyd-2 $H_2$-oxidizing activity and consequently restore overall $H_2$ production. Strain HDK103 (Δ*hyaB* Δ*hycA-I*), which only synthesizes active Hyd-2 enzymes, was used to investigate whether DTT addition influenced the $H_2$-oxidizing activity of Hyd-2. $H_2$-saturated buffer was applied to the electrode, and cell suspension was added before the reaction was initiated with 15 mM fumarate as electron acceptor and the $H_2$ trace recorded (Figure 5B). The addition of 5 mM DTT is indicated but did not inhibit $H_2$ oxidation. Hence, DTT did not inhibit the $H_2$-oxidizing activity of Hyd-2, and restoration of overall $H_2$ production is caused otherwise.

It is noteworthy that a redshift of the CCCP spectrum in the presence of DTT was observed, where the absorbance maximum shifts from 381 nm to 405 nm (Figure 5C). The reaction of CCCP derivatives with thiols has been characterized [31] and showed that they are inactivated in the reaction in the presence of DTT. Taken together, this supports the observation made above that the observed effects of CCCP are not merely due to its protonophore activity but rather could be attributed to its interaction with thiols from cysteine residues, possibly those within the FHL complex. Cysteine residues are essential for electron transfer within the FHL complex by coordinating the metal cofactors. Recently, the number of free thiols was determined and was reduced in strains unable to synthesize a functional FHL complex [32]. However, an inactivation of CCCP by DTT and re-formation of the *pmf* by the cell can also not be excluded.

## 2.5. Effect of Monovalent Cations on FHL Reactions

CCCP has been described to collapse both the ΔpH and the transmembrane electrical gradient (ΔΨ) component of the *pmf* [33]. It has been observed that FHL is more active and the *hyc*-genes

more strongly expressed under acidic conditions [34,35]. While a $\Delta$pH collapse would increase the availability of protons to the hydrogenase active site in the cytoplasm, and hence increase substrate availability, the $\Delta\Psi$ collapse represents reduction of the outer positive charge and inner negative charge of the membrane. In order to investigate the possibility that the apparent *pmf* dependence could be due to the $\Delta\Psi$ collapse mediated by CCCP, we analyzed the effects of cation addition on $H_2$ production.

Kinetic experiments on the electrode showed reduction of the $H_2$ production rate when increasing concentrations of $Na^+$ ions were present. The rates were reduced by 27% in strain CP734 in the presence of 200 mM sodium compared to in its absence (Figure 6A). When 100 μM CCCP was added to any of the reactions, the reduction was between 40%–50%, regardless of the sodium ion concentration present (Figure 6A, grey dots). Therefore, the presence of NaCl did not enhance the effect of CCCP in this strain.

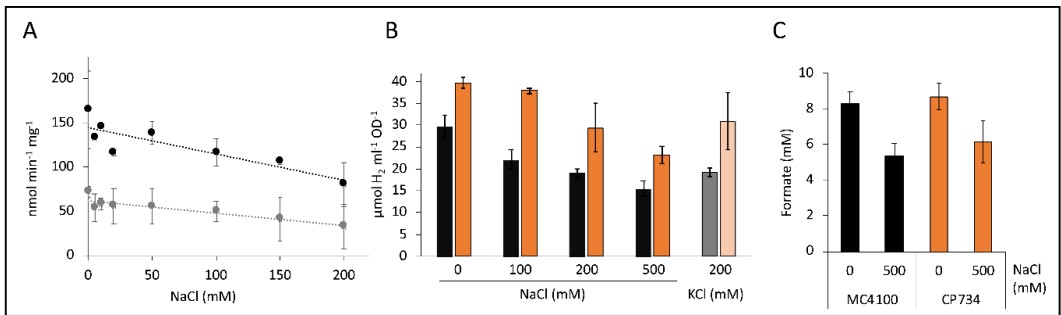

**Figure 6.** Effect of cations on $H_2$ production. Panel (**A**) shows the kinetic $H_2$ production rates of CP734 cells in the presence of different concentrations of NaCl (black dots) and in combination with 100 μM CCCP (gray dots). Reactions were initiated with 50 mM formate. Each point represents the average of two independent biological experiments with the respective standard deviation. The trend lines were calculated using linear regression. Panel (**B**) shows the $H_2$ content in the headspace of a 7 mL culture tube of MC4100 (black graphs) or CP734 (*ΔhyaB ΔhybC*, orange graphs) after anaerobic overnight growth in M9-minimal medium with 0.8% (*w/v*) glucose in the presence of NaCl or KCl as indicated. (**C**) The formate concentration of cultures from panel B was quantified using HPLC, as described in the Materials and Methods section. All values represent the average of three independent biological replicates and their standard deviations.

The net charge of the ions remains unaltered through the addition of NaCl or KCl, because the chloride counterion is present. However, the cation/$H^+$ antiporters counteract the increased salt concentrations that challenge the cell and concomitantly excrete excess cations, which is accompanied by an increased $H^+$ influx. It was observed that $\Delta\Psi$ decreased over the first 15–20 min after addition of KCl or NaCl [36], hence within the time-frame where $H_2$ production was monitored. Apparently, it is this disturbance of the electrical gradient that is detrimental to $H_2$ production by FHL.

Similarly, when cells were already grown in the presence of different concentrations of $Na^+$ or $K^+$ ions and subsequently the $H_2$ content in the headspace sampled by gas chromatography (GC), which indirectly reflects the activity of the FHL complex, a reduction in the presence of increasing concentrations of $Na^+$ ions could be observed (Figure 6B). Strain CP734 (*ΔhyaB ΔhybC*) showed an increased $H_2$ accumulation by 30%–70% compared to its parent MC4100 under all conditions. This effect is due to the absence of Hyd-1 and Hyd-2 enzymes, which oxidize part of the produced $H_2$. The $H_2$ content of strain MC4100 was reduced by 36% and 48% in the presence of 200 mM and 500 mM NaCl, respectively, while strain CP734 showed a similar reduction of 26% and 42% in the presence of 200 and 500 mM NaCl, respectively. This further supports the notion that $H_2$ oxidation, and hence the activity of the $H_2$-oxidizing enzymes, was not increased in response to NaCl. To verify whether the effect of $Na^+$ ions was specific, 200 mM $K^+$ ions were used instead and showed an identical reduction of the $H_2$ content by 35% and 22% for MC4100 and CP734, respectively. Therefore, the reduction was

not specific for a particular cation. However, $Na^+$-transport was shown to be $K^+$ dependent in *E. coli*, albeit not by direct exchange but rather is $H^+$-antiport coupled [37]. This interconnection of the two cation transport systems could explain the similar effect of both cations.

Headspace $H_2$ derives from supplemented glucose given to the cells during growth, which is converted to formate as one of the products of mixed-acid fermentation. Elevated concentrations of 500 mM $Na^+$-ions reduced the amount of produced $H_2$ (Figure 6B). In order to identify the fate of the carbon, the formate concentration was determined. The cultures had reduced formate amounts by 35% and 29% for MC4100 and CP734, respectively (Figure 6C). Other key metabolites of the mixed acid fermentation like succinate and acetate were similarly reduced under high salt conditions (data not shown). At the same time, the growth of the cultures was reduced by only 15% and was therefore not the main reason for the reduced formate production. This result indicates that under high salt conditions formate is not accumulated because of FHL inhibition, but rather, more generally, the entire mixed-acid fermentation was reduced.

## 2.6. The $Na^+$ Ionophore EIPA Enhances CCCP Inhibition

The sodium concentration of the cell is maintained by $Na^+/H^+$ antiporters that reside in the cell membrane and also function in regulation of the intracellular pH. *E. coli* synthesizes three MRP-type antiporter proteins (NhaA, NhaB and ChaA), which are responsible for $Na^+/H^+$ exchange [38]. Intriguingly, this class of $Na^+/H^+$ antiporters is homologous to the HycC protein of the FHL membrane domain and to the NuoLMN (*E. coli* nomenclature) proteins, the proton-pumping subunits of the NADH-oxidizing complex I [39]. Some of these antiporters, especially those homologous subunits within the membrane domain of respiratory complex I, can be inhibited by EIPA (5-(*N*-ethyl-*N*-isopropyl)-amiloride) [40]. *Klebsiella pneumoniae* complex I was shown to translocate $Na^+$ ions instead of protons and some evidence for $Na^+$ translocation of *E. coli* complex I exists, but this might merely be a secondary effect [41,42]. In addition, in thermophilic organisms like *Thermococcus onnurineus*, energy conservation functions in an intricate mechanism that couples formate oxidation with concomitant $H_2$ production for the generation of a proton gradient. This proton gradient is subsequently converted into a $Na^+$ gradient by a $Na^+/H^+$-antiporter and then employed by a $Na^+$ ATPase for ATP synthesis [43].

Due to the observed reduction of $H_2$ production by NaCl, we tested the effect EIPA had on the $H_2$ production of FHL and found that, on its own or combined with 200 mM NaCl, the effects were negligible as described before [8]. Surprisingly, however, when EIPA and CCCP were given shortly after each other, they were able to inhibit $H_2$ production completely (Figure 7A). A similar effect was also determined for strain CP734 ($\Delta hyaB$ $\Delta hybC$, Figure 7B); however, inhibition of $H_2$ production was not as complete as for MC4100. EIPA on its own also had no effect (Figure 7B). Notably, inhibition was also not complete for MC4100 when the reaction was started with formate instead of glucose (Figure 7C), and no effect was seen in strain CP734 ($\Delta hyaB$ $\Delta hybC$) when formate was used to initiate the $H_2$ production (Figure 7D). This indicates that the activity of the $Na^+/H^+$ antiporter plays a crucial role during $H_2$ production from glucose. However, the MRP-type membrane subunit HycC of the FHL complex, which is essential for the reaction, is not inhibited by EIPA. Taken together, this could imply that a $H^+$ gradient is converted into a $Na^+$ gradient at the cell membrane during catalysis and that the FHL reaction couples both. Further experiments will be necessary to find the target of the ionophores.

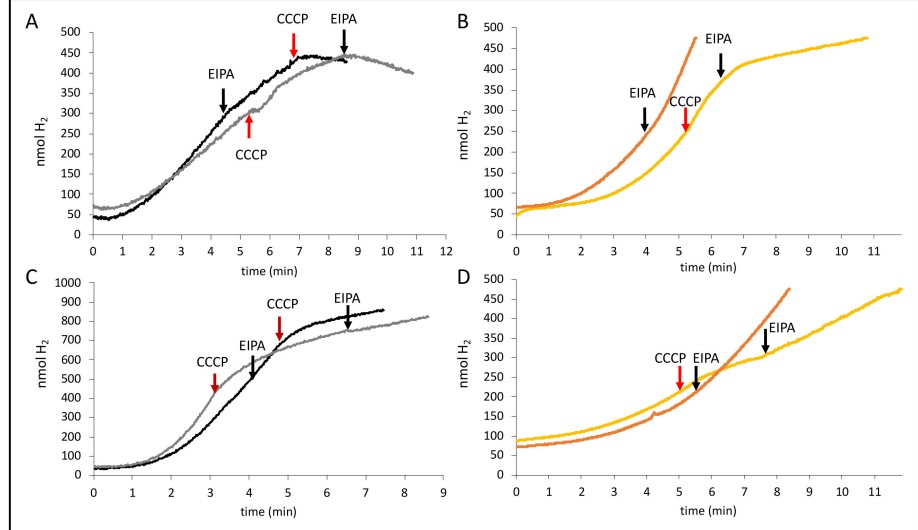

**Figure 7.** Effect of EIPA combination with CCCP. Cells of MC4100 (black, panels (**A**) and (**C**)) or CP734 (orange, panels (**B**) and (**D**)) were grown anaerobically in TGYEP, pH 6.5, and applied to the electrode as described in the Materials and Methods section. The reaction was initiated with glucose (top panels (**A**) and (**B**)) or formate (bottom panels (**C**) and (**D**)). Where indicated, 50 μM CCCP (red arrows) or 50 μM EIPA (black arrows) were added to the reaction.

## 3. Materials and Methods

### 3.1. Strains and Growth Conditions

Strains are listed in Table 3. Strains CP630 and CP631 were constructed using P1*kc*-mediated phage transduction of the Δ*hyaB* and Δ*hybC* alleles of the Keio-collection strains JW0955 and JW2962, respectively, as described [44]. Strains were routinely grown aerobically in LB medium or on LB agar plates at 37 °C. For analysis of FHL activity, the strains were grown anaerobically as standing liquid cultures for 16 h in 50 mL TGYEP medium, pH 6.5 (1% (*w/v*) tryptone, 0.5% (*w/v*) yeast extract, 0.1 M potassium phosphate buffer, pH 6.5, 0.8% (*w/v*) glucose) at 30 °C [45]. Cultures for $H_2$ headspace and HPLC analyses were grown in phosphate-buffered M9-minimal medium supplemented with 0.8% (*w/v*) glucose as described [46]. When indicated, sterile solutions of NaCl or KCl were added. CCCP was used at 100 μM unless otherwise indicated. Cultures were harvested by centrifugation at 4000× g for 15 min at 4 °C, resuspended in 800 μL 50 mM Tris/HCl, pH 7.0, and kept on ice until further use.

**Table 3.** Strain list.

| Strain | Genotype | Reference |
|---|---|---|
| MC4100 | F⁻ *araD139* Δ(*argF-lac*)U169 λ *rpsL150 relA1 deoC1 flhD5301* Δ(*fruK-yeiR*)725(*fruA25*), *rbsR22*, Δ(*fimB-fimE*)632(::IS1) | [47] |
| CP630 | Like MC4100, but Δ*hyaB* | This study |
| CP631 | Like MC4100, but Δ*hybC* | This study |
| CP734 | Like MC4100, but Δ*hyaB* Δ*hybC* | [24] |
| HDK103 | Like MC4100, but Δ*hyaB* Δ*hycA-I* | [48] |

### 3.2. Enzymatic Assays

The kinetic activity of the FHL complex was determined on a modified Clark-type electrode equipped with an OXY/ECU module (Oxytherm, Hansatech Instruments, Norfolk, UK) to reverse the polarizing voltage to −0.7 V. A volume of 2 mL degassed 50 mM Tris/HCl buffer, pH 7.0, at 30 °C was added to the chamber prior to adding 50 μL of cell suspension. The reaction was started either with 50 mM formate or with 14 mM glucose. When the effect of sodium ions was studied, the ammonium salt of formate was used to initiate the reaction, which increased the pH by 0.02 units. The amount

of $H_2$ was calibrated as described [49]. $EC_{50}$ values were calculated with Origin Pro 2017G software (OriginLab, Northampton, MA, USA). The protein content of the respective cell suspensions was determined using the method of Lowry in a micro-scale assay [50]. Optical densities and spectra were recorded with a Tecan plate reader.

The $H_2$ content of the gas headspace of a 15-mL Hungate tube filled with 7 mL of culture was determined using gas chromatography with a GC2010 Plus Gas Chromatograph (Shimadzu, Kyōto, Japan) as described [39].

For the reverse FHL reaction, the cells were initially grown anaerobically in TGYEP, pH 6.5, for 16 h at 30 °C, harvested and resuspended in 50 mM Tris/HCl, pH 7.5. Cell suspensions were further mixed with different concentrations of CCCP, as indicated, and incubated under 2 atm pressure of $H_2$ and $CO_2$ 1:1 mixture for 16 h at 37 °C. Cells were then removed by filtration through a 0.2 µM filter and supernatants applied to an HPLC system equipped with an Aminex HPX-87H column (Bio-Rad, Hercules, CA, USA), and formate concentrations were determined as previously described [8]. The culture supernatant was centrifuged and subsequently passed through a 0.2 µM filter prior to loading onto the HPLC system.

Calorimetric assays of FdhH activity and hydrogenase activity were carried out with crude extracts from anaerobically grown cells. The cells were harvested, sonicated for 2 min at 20 W with 0.5 s pulses and briefly centrifuged. Anaerobic cuvettes were prepared containing 0.8 mL 50 mM Tris/HCl buffer, pH 7.0, and 4 mM benzyl viologen. The FdhH reaction was started with 15 mM formate, and the hydrogenase reaction was started by adding crude extract to the cuvettes after exchange of the $N_2$ headspace with $H_2$. The signal at 600 nm was recorded, and an $E_M$ value of 7400 $M^{-1}$ $cm^{-1}$ was assumed for reduced benzyl viologen. Data were derived from three independent biological replicates. Protein concentration was determined using the Lowry method [50].

## 4. Conclusions

The data presented here confirm CCCP-dependent inhibition of FHL activity. Here, we quantified this effect for the first time and saw striking differences depending on the presence of active $H_2$-oxidizing Hyd enzymes. The CCCP-dependent inhibition showed clearly that FHL did not contribute to proton translocation across the membrane, and in contrast, the data suggest it is driven by proton influx. Nevertheless, the data further highlight that the effect of CCCP might not be due to its protonophore activity but might rather be indirect, either by interacting with thiol groups within the complex or by disturbing the charge distribution at the membrane. The absence of an effect of another protonophore on the $H_2$ production further supports this finding. The sodium/potassium inhibition of the $H_2$ production showed that in order for it to function most effectively, the FHL complex requires low external cation concentrations. Our data clearly suggests that $H_2$ production couples $H^+$ influx with $Na^+$ efflux. However, evidence shows that it is not the MRP antiporter subunit HycC of the FHL complex that is directly involved, but rather it is the cation/$H^+$ antiport of the membrane that influences $H_2$ production.

**Author Contributions:** C.P. conceived the study; J.T.M. and C.P. collected and analyzed the data, C.P. wrote the original draft and supervised J.T.M. All authors have read and agreed to the published version of the manuscript.

**Funding:** We acknowledge the financial support within the funding program Open Access Publishing by the German Research Foundation (DFG).

**Acknowledgments:** We are grateful to Julia Fritz-Steuber (University of Hohenheim, Germany) and Etana Padan (Hebrew University of Jerusalem, Israel), who kindly provided strain EP432. We thank Gary Sawers (Martin-Luther University Halle-Wittenberg, Germany) for fruitful discussions and support in writing this manuscript.

**Conflicts of Interest:** The authors declare no conflict of interest.

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
