# Peer review of "Susceptibility of the Formate Hydrogenlyase Reaction to the Protonophore CCCP Depends on the Total Hydrogenase Composition"

_inorganics, doi:10.3390/inorganics8060038_

Round 1
Reviewer 1 Report
Overall:
The possibility that CCCP directly affects the enzymatic activities of the FHL complex remains to be eliminated to the end, which seems to be crucial to address the main issue of the paper.
I suppose that this can be resolved by the FDH/H2ase activity assays with the purified enzymes (the FHL complex and other hydrogenases) in the presence and absence of CCCP. The authors may be able to cite the data from the previous work, or, if it does not exist, can perform the experiment as they have already established in vitro assay in the reference [8].
Other comment:
Line 206
Figure 4 should be Figure 3.
Author Response
Dear reviewer 1, please find our response in the attached pdf file.

Reviewer 2 Report
The researchers use whole cell assays to monitor the rate of Hydrogen production by the FHL complex of E. coli and its inhibition by the addition of inhibitors of proton or ion transport. Comparisons are made with engineered strains lacking other hydrogenase enzymes (Hyd-1, Hyd-2 or both) which primarily act to consume H2 produced by the FHL complex under the assay conditions. Inhibition by CCCP initially suggest a link to the cellular pm fie H2 production is drive by the protonmotive force, but closer analysis suggests an indirect inhibitory effect. Further experiments suggest Hydrogen production may be indirectly linked to proton influx via the efflux of Na cations. There is clearly still much to learn about this complexities of the FHL complex and its role in fermentative growth but this paper is a strong foundation for further study. In my opinion it represents an important next step in unravelling a particularly intriguing question as to the biochemical function of the FHL complex, its role in bacterial hydrogen metabolism and its integration into wider bacteria physiology.
I have the following suggestions/queries for the authors
Intro
The manuscript assumes a lot of prior knowledge about the particulars of this system, I would suggests an introductory figure, a schematic of the complexes showing their orientation relative to the membrane and each other, and an indication of hydrogen cycling in the cell would be a valuable addition to set the scene for those less familiar with the subject area.
Line 55 reaction directions should be stated explicitly for clarity ie change forward to H2 production and reverse to formate production.
Results and discussion
Could the authors clarify how the inhibition curve (figure 2b) particularly slope ratios were calculated? For example the text states there is no change in slope with the addition of 10 µM CCCP and figure 5a shows more or less a straight line. Yet the slope ratio in panel b is shown as 0.9. The highest concentration of CCCP (400 µM) yields a negative slope after CCCP addition yet the ratio is positive.
Also are the values in panel b averages of multiple experiments? If so, how many should be stated and error bars added.
Line 170-172. The activity of Hyd-1 is stated to be lower than Hyd-2, is this Hydrogen oxidation activity? Could the authors provide a reference for this? My understanding was H2 oxidation activity was similar between Hyd-1 and Hyd-2. Is it possible to quantify the amount of Hyd-2 in membranes (Western Blotting?) for the various strains to test the assertion the Hyd-2 levels are elevated in CP630?
In figure 5a are the kinetic experiments shown averages of independent experiments? If so how many and show error bars. How was the trend line calculated? Showing a trend line for the +CCCP samples would also be helpful.
In the EIPA inhibition experiments (figure 6) panels A and B are for reactions initiated with glucose, while C and D are initiated with Formate. In lines 346-347 the authors state inhibition is only observed when the reaction is started with glucose, yet the curves In panel C (MC4100 and formate) look very similar to panel B (CP374 and glucose), with the latter only being attributed to inhibition. The authors should address the discrepancy.
Methods – Does addition of ammonium formate have any effect the pH of the assay?
Was the tris buffer pH’d at 30 °C prior to experiments, otherwise the final assay pH will be lower than expected?
Author Response
Dear reviewer 2, please find our response in the attached pdf file.

Reviewer 3 Report
In the current manuscript J.K. Marloth and C. Pinske investigated the H2 evolution and the formate production of E. coli strains synthesizing a formate hydrogenlyase (FHL), an enzyme that couples formate oxidation to H2 production. The authors observed that the H2 evolution of FHL was highly affected by the presence of CCCP protonophore, while the formate production was enhanced.
The authors correlated the CCCP effect on the presence of the E. coli [NiFe]-hydrogenase Hyd-1 and Hyd-2 using different strains containing i) genes encoding for the two hydrogenases (MC4100), ii) single deletion of Hyd-1 (CP630, ΔhyaB) or Hyd-2 (CP631, ΔhybC) and iii) complete deletion of the two hydrogenases (CP734, ΔhyaB ΔhybC).
After careful evaluation of the manuscript, I do not recommend publication of this study in the MDPI Inorganics journal.
The present study is limited in originality as the authors reported on the CCCP effect on the formate production and formate-dependent H2 evolution in their previous study (ref. 8 of the current manuscript, Microbiologyopen 2016, 5, 721–737). Only an incremental work with preliminary data is presented, using two new strains CP630 and CP631 having single deletion of the large subunit genes encoding for Hyd-1 and Hyd-2 proteins. Besides, some discrepancies raised comparing the two studies. Indeed, in the current study using the strain CP734 (ΔhyaB ΔhybC) the formate-dependent H2 evolution proved not affected by CCCP chemical (Fig. 1). On the contrary, in the previous study (ref. 8 Table 4) only 58% of the H2-evolution rate was retained.
Furthermore, maximal activity of CP734 strain was 37.8 ± 7.6 nmol H2*min-1 *mg-1 in Microbiologyopen 2016, 5, 721–737, while in this study the activity is 3 times higher. Did the author change cultivation procedure, chemical composition and or pH of the assay?
I suspect that the difference of the activity might be related to the amount of formate added. Indeed, formate concentration was 15 mM in the previous study while here has been increased to 50 mM (ca three times higher). Please normalize H2 evolution data to the formate concentration to compare previous CCCP inhibition experiments with the current data.
Additional major concerns:
1) The H2-production rates are measured on resting cells. I recommend the normalization of the activity to the cell optical density. Otherwise, data reported in different figures cannot be directly compared. As an example, in Figure 1B H2 evolution of CP630 (ΔhyaB) strain was ca 200-230 nmol after 2 min, while in figure 2A the activity is half of that reported in Figure 1B.
2) The authors observed (lines 118-128, Table 1) that the H2 evolution of the CP631 strain (ΔhybC) looks like the H2 production of CP734 (ΔhyaB, ΔhybC) strain, while CP630 (ΔhyaB) strain´s activity looks very similar to the parental strain MC4100.
Considering the reported data, the H2 evolution is affected mainly by the Hyd-2 deletion (see Table 1). The authors experiments do not support any effect on the FHL H2 evolution driven by the deletion of Hyd-1 (CP630, ΔhyaB).
3) While the qualitative effect of the CCCP inhibitor can be observed in figure 2A, it is hard to quantify the real EC50 values because the ratio CCCP/cells has not be considered. An alternative way to perform the experiment would be to do incremental CCCP additions on the same cell resuspension mixture, monitoring the change of the H2 rate at each addition. Please, reconsider the data in Table 2 performing data analysis based on the real ratio CCP/resting cells and provide more biological replicates to have a standard deviation.
4) The experiments described in paragraph 2.4 needs to be carefully evaluated and additional experiments are required. Indeed, the DTT treatment clearly induced some chemical modifications to the CCCP inhibitor (See figure 4C) and the recovery of the activity might be ascribed to a CCCP decomposition. A clear control experiment is missing, and I would suggest to preincubate CCCP and DTT before injection in the reaction mixture. If inhibition would not be detected in the control experiment, it could validate a DTT-mediated decomposition of the CCCP compound. In addition, TCEP (a thiol-free reducing agent able to reduce cysteine S-S bonds) should be tested and its effect on CCCP evaluated.
5) Figure 2B is missing error bars. Is the EC50 calculated on a single experiment with different CCCP concentrations? Please provide at least three replicates, possibly following suggestions in comments 1 and 3.
6) Experiments in paragraph 2.6 on the CCCP/EIPA effect on the glucose-mediated H2 evolution are too preliminary to allow any conclusion. Why the activity on CP734 strain is not abolished completely as in the case of MC4100 strain? More experiments need to be performed.
Minor points:
1)Page 2, line 57-58. Add pH 7.0 at the redox potentials in brackets. Indeed, the standard redox potential per definition refers to pH 0. What the authors are reporting is the thermodynamic value at pH 7.0.
2) The data points presented in figure 3 cannot be distinguished. Please divide the graph in two distinctive sections: a) 1-75 µM of CCCP and b) 100-500 µM of CCCP
3)Data in Figure 5A are missing error bars.
Author Response
Dear reviewer 3, please find our response in the attached pdf file.

Round 2
Reviewer 3 Report
J.K. Marloth and C. Pinske submitted a new version of their manuscript enriched with additional figures and control experiments supporting their previous assumptions. Furthermore, considering that the previous published work has been carried out in another lab using different instruments and protocols (e.g. formate concentration), some discrepancies between the two studies (i.e. enzymatic activity) are explainable.
I´m glad to see that the authors performed important control experiments missing in the first version of the manuscript and overall:
1) DTT/CCCP preincubation (Fig. 5a), where the authors could show that CCCP was unable to inhibit H2-production when it was already pre-incubated with DTT. This leaves open a DTT-mediated inactivation of CCCP and re-formation of the pmf by the cell.
2) The absence of CCCP inhibition of the hydrogenase and formate dehydrogenase half reaction when crude extracts were assayed using benzyl viologen dye (lines 326-331).
The revised version includes a new Figure 1, which introduces the experimental work of the paper. Figures 4, 5 and 6 have been modified as requested also by other referees and now include new data, biological replicates, and an increased readability.
Considering the modifications, additional changes proposed by other referees including new references, and some self-criticism of the authors that smoothed their comments on the CCCP inhibition effect, redefining their EC50 as an apparent half-maximal inhibition, this paper is greatly improved from the prior version and might serve as starting point to unravel new aspects on the FHL function. In particular, the EIPA/CCCP results are very puzzling and should be targeted for future research studies.
Minor points:
Line 81: “Further details see text” should be replaced with “Further details are highlighted in the main text.
Line 611: Please report the concentration of Tris Buffer.
Line 616: Protein concentration was determined by the Lowry “method”.
Author Response
Thank you for your kind assessment of the revised manuscript.
There were no major points to be addressed. The following minor points have been changed according to the reviewers suggestions:
Line 81: “Further details see text” should be replaced with “Further details are highlighted in the main text.
Answer: line 53 - done
Line 611: Please report the concentration of Tris Buffer
Answer: the concentration is written in the materials and methods section as 50 mM (lines 114, 394, 401 and 414; the document has no line 611) and this has also been added to the Tris buffer mentioned in line 424.
Line 616: Protein concentration was determined by the Lowry “method”.
Answer: Line 429 has been changed to 'the Lowry method', otherwise this would have been very spectacular.